# Rock-Breaking Characteristics of High-Pressure, Dual-Stranded Water Jets

Yue Pan, Shengyu Zhai, Kangchao Pei *, Hao Yuan and Fulin Huo

School of Mechanical and Equipment Engineering, Hebei University of Engineering, Handan 056038, China; 18237378373@163.com (S.Z.)
* Correspondence: peikc378312131@163.com

**Abstract:** Because of the unclear understanding of the characteristics associated with coupled rock breaking using multiple water jets, a numerical model combining smoothed particle hydrodynamics (SPH) and the finite element method (FEM) was established to investigate the rock-breaking capacity of a high-pressure, double-stranded water jet structure. The effectiveness of this model was verified through field experiments. The study further examined the specific energy required for rock breaking using the high-pressure double water jets and analyzed the effects of jet pressure, nozzle diameter, jet impact angle, and impact point spacing on rock-breaking volume. The results demonstrate that the rock-breaking ability of a high-pressure double water jets is better than that of a single water jet. When the impact angle of the high-pressure double water jets was 15° and the distance between impact points was 2.0 d, the rock damage effect was the best. By comparing the specific energies for rock breaking of a single water jet and a double water jet, it was concluded that the best rock-breaking nozzle diameter is 1.6 mm. Furthermore, an orthogonal testing approach was employed to determine the main and secondary factors influencing the rock-breaking energy of the high-pressure double water jet. The order of significance was found to be jet pressure > impact angle > impact point spacing > nozzle diameter. These findings provide valuable guidance and reference for application in the coal mining industry.

**Keywords:** high-pressure double water jet; specific energy for rock breaking; jet flow angle; optimal parameters

## 1. Introduction

As a result of its advantages of being nonsparking, dust-free, low heat and low vibration [1,2], high-pressure water jet breaking technology has broad application prospects in underground coal mines to assist with rock breaking or to replace the mechanical cutting of coal rock. Improving the jet pressure and changing the jet structure can improve the rock-breaking capacity of the jet [3,4]. The existing single-nozzle rotating double-jet structure and single-nozzle multihole jet structure are mainly used in drilling operations in the oil and coal industries, and the rock-breaking efficiency of the multihole nozzle is much higher than the rock-breaking efficiency of the combination rotary direct jet and rotary jet [5].

Oh et al. [6] investigated the granite fracture characteristics by varying the geometric parameters of water jets in a high-pressure water jet system. Based on the numerical model of SPH, Xue et al. [7,8] simulated the propagation of stress waves in rocks under the action of pulsed water jets, and the study demonstrated that the stress propagation rules and fracture characteristics are the keys to identifying the mechanism of water jet impact rock fragmentation. Liang et al. [9] found that in the early stage of the high-pressure water jet's impact on coal rock, the dominant damage is the impact of the water jet's dynamic load, and the quasistatic damage to the coal rock after the jet impact's stabilization is lower. Wu et al. [10] used the Voronoi method to create random polygonal particles to approximately simulate the microstructure of rocks, and they found that the rock breakage

performance under water jet action is significantly affected by grain size, irregularity, ductility, microstrength and microparameter heterogeneity. Lu et al. [11] studied the dynamic evolution characteristics of the fluid structure of the truncated pulsed water jet, tested the rock-breaking ability of the truncated pulsed water jet, and compared it with the conventional cylindrical water jet in terms of its smooth structure and rock-breaking ability. Tripathi et al. [12] compared the erosion performance of a continuous water jet and pulsed water jet on rock, and they concluded that its influence on the granite erosion performance was reflected in the change in the process parameters. Liu et al. [13] studied the formation process of impact pits and slits on rock in addition to the velocity and distribution of the water jet, and they revealed the rock fracture mechanism under the action of the water jet using the coupled Euler–Lagrange method and the finite-discrete element method. Sheng et al. [14] observed sandstone and shale after high-pressure water jet impact via computed tomography (CT) and electron microscopy, and they found shale in difficult-to-occur volume fragmentation caused by water jet impact. Ge et al. [15,16] analyzed the damage fracture characteristics of coal, sandstone, and shale using scanning electron microscopy (SEM), nuclear magnetic resonance imaging (NMRI), and CT techniques, and they found that the coal rocks were mainly fractured from the water wedge action in the longitudinal direction, the sandstones were "spindle-shaped" fracture pits caused by exfoliation damage, and the shales were fractured from the water wedge action and stress waves in the shallow lateral direction. The shale showed a shallow transverse annular fracture and longitudinal fracture at the bottom caused by the combined action of water wedging and stress waves.

The aforementioned studies demonstrate the various forms and mechanisms of high-pressure water jet rock breaking from different angles and methods. However, previous research has only analyzed single-nozzle, single-hole water jets impacting on rocks or studied single-nozzle, multihole water jet rotating boreholes impacting on rocks. The characteristics of coupled rock breaking using multiple water jets remain unclear. Therefore, a single-factor experimental analysis of high-pressure double water jets was conducted via a mutually verifying experiment and simulation. An equivalent water injection method was employed to measure the rock crushing volume and to calculate the specific energy consumed to crush rocks. Finally, a series of test parameters were selected for an orthogonal test analysis to determine the optimal fracture parameters of the rock.

## 2. Numerical Simulation

### 2.1. Model Building

A high-pressure impact dynamics problem involving many variables, including both the high-pressure flow impact of the water jets and the stress wave response created by the rock under this effect, is the progressive fracturing of rocks by high-pressure water jets. Given the difficulty of using FEM to calculate massive deformation problems, SPH has the inherent flaws of tensile instability, difficulty enforcing boundary conditions, and low computing efficiency [17]. Because of the advantages of large deformation under SPH and high computational accuracy and efficiency under FEM, the coupled SPH-FEM algorithm was used here to simulate the process of rock fracture fragmentation. This allowed us to not only obtain greater precision in rock damage characteristics but also to better simulate the large instantaneous deformation and high strain rate in the impact process.

The SPH-FEM contact coupling algorithm calculates the contact force between the SPH particle and the finite element using the potential contact gradient and point and surface contact. The point–surface contact algorithm was utilized in this paper to determine the coupling between the water jet and the rock interface. The slave nodes are SPH particles, and the control parameters are the node number, spatial position, and mass; the primary surface is described as the rock element of the finite element description. The meshless contact technique is reflected in the implementation of the contact force between the SPH particles and the finite element.

In practice, the water jet shape is dispersed, and a radial velocity exists in addition to a small number of macroscopic and microscopic defects inside the rock. In this paper, to simplify the actual problem the following model was assumed: the process of a high-pressure water jet impacting on rock was simplified to a time-limited process of a rectangular water beam impacting on rock, where the jet was a single-phase homogeneous fluid, and the radial velocity of the jet was not considered; the rock was a homogeneous material, and the effect of initial cracks and pores was not considered. The geometric model of the rock was 80 mm long and 60 mm wide, and BOUNDARY_SPC_SET was used to constrain the bottom motion of the model; BOUNDARY_NONREFLECTING was used to make the sides and bottom of the model a transmissive interface to eliminate the influence of the boundary on the damage. The model area was divided into a fluid domain and a solid domain, with the rock being the solid domain. The mapping division method was used to divide the rock into 120,000 thin hexahedral units, and the geometric model is shown in Figure 1.

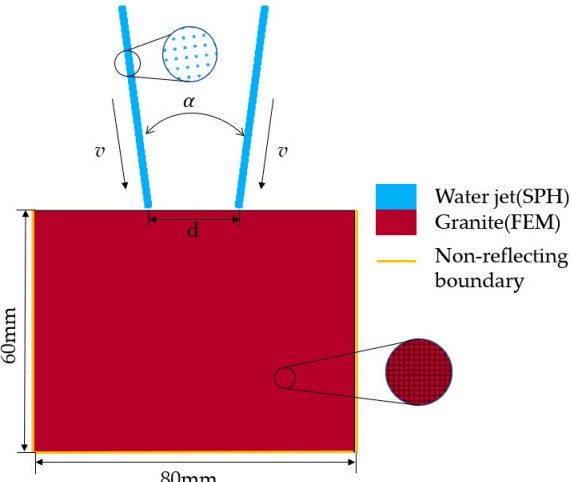

**Figure 1.** Two-dimensional model of high-pressure double water jets impacting rock.

### 2.1.1. Water Jet Model

The water jet was the fluid domain, and the fluid domain was directly defined by dividing the 4000 SPH particles and 2000 SPH particles per water jet. The water jet per share was continuous with a diameter of 1.0 mm and an impact velocity of 300 m/s. The Grüneisen equation of state [18] was used in the simulation of water jets to model the pressure in the impact state of the water jets, with the expression as follows:

$$P = \frac{\rho_0 C^2 \theta \left[1 + \left(1 - \frac{\gamma_0}{2}\right)\theta - \frac{a}{2}\theta^2\right]}{\left[1 - (S_1 - 1)\theta - S_2 \frac{\theta^2}{\theta+1} - S_3 \frac{\theta^3}{(\theta+1)^2}\right]^2} + (\gamma_0 + a\theta)E \tag{1}$$

where $\rho_0$ is the initial density of water, and $C$ and $\theta$ are the speeds of the sound and volume strain, respectively; $\theta = (\rho/\rho_0 - 1)$; $\gamma_0$ is the Grüneisen coefficient; $a$ is the first-order volume correction to $\gamma_0$; $S_1$, $S_2$, and $S_3$ are the material constants of water; and $E$ is the initial internal energy.

The parameters of the constitutive model for water are shown in Table 1.

**Table 1.** Parameters of the Grüneisen equation of state for water.

| $\rho_0/(\text{g·cm}^{-3})$ | $C/(\text{m·s}^{-1})$ | $\gamma_0$ | $a$ | $S_1$ | $S_2$ | $S_3$ |
|---|---|---|---|---|---|---|
| 1.05 | 1480 | 0.5 | 0 | 2.56 | −1.986 | 1.227 |

### 2.1.2. Rock Material Model

In this paper, the Johnson–Holmquist II constitutive model for brittle materials was utilized to characterize the nonlinear impact dynamics of rocks under water jet penetration, aimed at providing a more accurate description of the rock breakage process [19,20].

An accurate description of the high-pressure equation of state should be used in determining rock damage. Therefore, the polynomial equation of state herein describes the correlation between hydrostatic pressure, $P$, and volumetric strain, $\mu$. The corresponding expressions are provided as follows:

$$P = K_1\mu + K_2\mu^2 + K_3\mu^3 + \Delta P \tag{2}$$

where $K_1$ is the bulk modulus, $K_2$ and $K_3$ are constants, $\mu$ is the volumetric strain, $\Delta P$ is the increment of pressure beyond the hydrostatic pressure. The JH-2 model's material parameters for rocks are provided in Table 2.

**Table 2.** Material parameters for rocks.

| $\rho/(\mathbf{kg/m^3})$ | $K_1$/**GPa** | $K_2$/**GPa** | $K_3$/**GPa** | *Hel*/**GPa** | *T*/**GPa** | *G*/**GPa** |
|---|---|---|---|---|---|---|
| 2657 | 55.6 | −18 | 3980 | 4.5 | 0.054 | 28 |
| A | N | M | B | C | $D_1$ | $D_2$ |
| 0.7 | 0.56 | 0.61 | 0.68 | 0.005 | 0.05 | 0.8 |

### 2.2. Simulated Analysis

LS-DYNA R11.1 software was used to establish a two-dimensional numerical model of the high-pressure, dual-stream water jet impacting on rock. The two-dimensional model is shown in Figure 1. Figure 2 depicts the time sequence diagram of the microcrack propagation induced by a double water jets with an angle of incidence of $\alpha = 0°$. As can be seen from Figure 2, there was an obvious central crack inside the rock at 35 μs. At 60 μs, the layered crack and radial crack in the rock showed a significant extension trend, and the macro-fracture behavior appeared at the impact point. At 70 μs, two V-shaped crushing pits appeared at the impact point of the water jet; meanwhile, the layered cracks intersected and connected with each other. During the last 70–120 μs, the fracture pit continued to expand, and the layered crack reached its maximum extent. The damage was mainly concentrated at the bottom of the fracture pit. The original radial crack transmitted into the layered crack, and then extended and expanded to the middle area of the two impact points; the cuttings in the middle of the two water jet impact points were removed, and the crushing pit was connected.

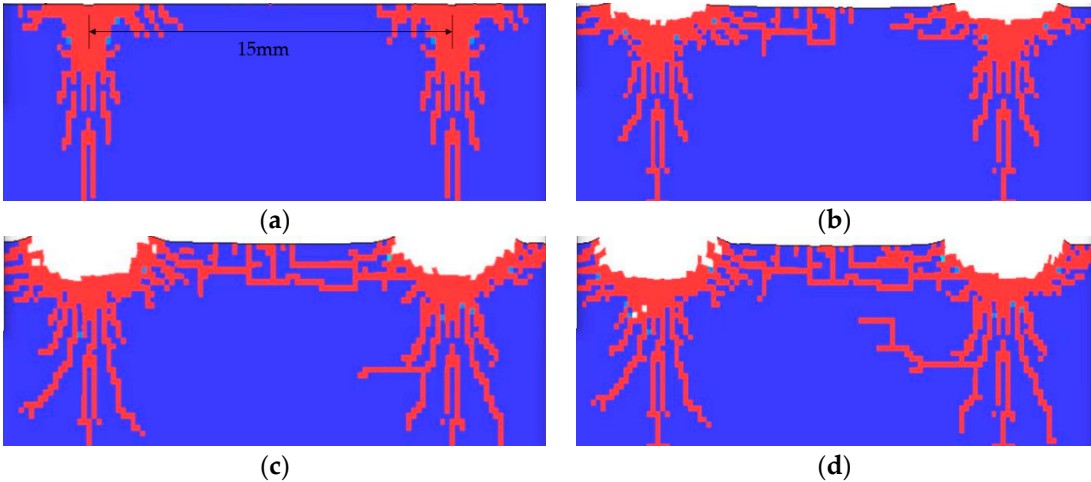

**Figure 2.** Sequence diagram of crack propagation in rock: (**a**) 35 μs; (**b**) 60 μs; (**c**) 70 μs; (**d**) 120 μs.

LS-DYNA R11.1 software was used to conduct a numerical simulation and analysis of the high-pressure double water jets impacting on rock. It was found that when the high-pressure double water jets impinged on rock with a pressure of 50 MPa, the rock specimen with a uniaxial compressive strength of 80 MPa could be broken through, and the main broken rock area was between the impact points of the two water jets. The main failure mode was shearing failure under the coupling action of two high-pressure water jets. The rock underwent compression caused by shock waves, and then the rock began to release pressure, causing it to break under greater shear. The impact load between the two high-pressure water jets interacted with each other to accelerate and intensify the rock failure.

In conclusion, with the formation of initial fractures on the rock surface, the rocks in the impact region of two high-pressure water jets released compression energy laterally, which caused the rocks to exhibit shear failure.

## 3. Experimental Analysis

### 3.1. Specimen Analysis

The test in this paper used the high-pressure water jet testing platform at the Fluid Laboratory of Hebei University of Engineering. The parameters for each part of the equipment used were an (1) Italy AR high-pressure plunger pump with a rated pressure of 50 MPa, adjustable: 0~50 MPa, rated flow of 44 L/min, rated power of driving motor of 44 KW, and maximum speed of 1470 r/min. The whole pump group was composed of a three-phase asynchronous motor through elastic pin coupling and a high-pressure plunger pump connection. (2) Other components included a shock-proof pressure gauge of 0~100 MPa; pressure-regulating valve with a limit value of 55 MPa; D-type seal; ZG pipe thread seal; DN15 high-pressure hose; high-pressure two-way screw ball valve PN500; filter; high-pressure spray gun; and 0.8 mm, 1.0 mm, 1.2 mm, 1.4 mm and 1.6 mm high-pressure nozzle. (3) Two high-pressure spray guns were converted into dual-stream water jet rock-breaking test spray guns that could be used to adjust the jet angle and jet distance and as nozzle replacements.

The test schematic is shown in Figure 3. In the process of the high-pressure water jet impacting on the rock and according to Bernoulli equation, the water jet is a continuous jet:

$$\frac{p_0}{\gamma} + \frac{\alpha_0 v_0^2}{2g} = \frac{p_{\text{out}}}{\gamma} + \frac{\alpha_i v_i^2}{2g} + h_{ji} + h_{fi} \tag{3}$$

where $p_0$ is the inlet pressure, MPa; $v_0$ and $v_i$ are the velocity of the inlet or orifices, m/s; $\alpha_0$ and $\alpha_i$ are the kinetic energy correction coefficients, and if it is turbulent flow, then $\alpha_0 = \alpha_i \approx 1$; $\gamma$ is the product of the density of water and the gravitational acceleration, N/m$^3$; $p_{\text{out}}$ is the outlet pressure, and if atmosphere pressure is adopted, then $p_{\text{out}} = 0$; $h_f$ is the frictional resistance loss, MPa; and $h_j$ is the local resistance loss, MPa.

Two points inside and outside of the nozzle's outlet were selected, and the fluid continuity equation was applied, as shown in Equation (4). Assuming the nozzle's exit is a cylindrical cavity, $A = (\pi d^2)/4$; $A_1$ and $A_2$ are the cross-sectional areas inside and outside of the nozzle, mm$^2$; $\rho^1 = \rho^2 = \rho^3$; and $d$ is the nozzle's diameter, mm.

$$\rho_1 \cdot v_1 \cdot A_1 = \rho_2 \cdot v_2 \cdot A_2 \tag{4}$$

where $p_1 \gg p_2$, $(d_2/d_1)^4 \ll 1$, and the jet density $\rho = 998$ kg/m; after the vertical Equation (4), it can be concluded that $v_2$ and its relationship with $p_1$ is as follows:

$$v_2 = 44.7\varphi\sqrt{p_1} \tag{5}$$

where $\varphi$ is an empirical parameter related to the Reynolds number, ranging from 0.9 to 0.98.

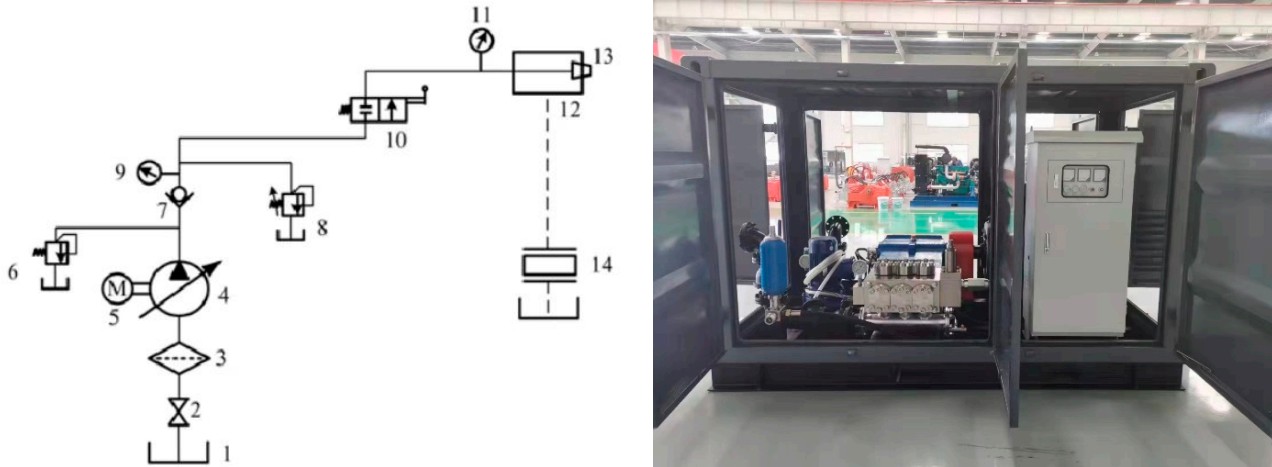

**Figure 3.** Schematic diagram of the test equipment: (1) water tank; (2) stop valve; (3) filter; (4) high-pressure pump; (5) motor; (6) safety valve; (7) check valve; (8) relief valve; (9) pressure gauge; (10) reversing valve; (11) pressure gauge; (12,13) testing platform; (14) recovery filter.

The rock specimens used in this paper were all cuboid blocks of 10 cm × 10 cm × 10 cm made from collected natural rocks. The prepared rock specimens were cored, and the obtained core end faces were polished so that the upper and lower end faces were parallel to each other and perpendicular to the core axis, and a cylinder with a diameter $D$ = 50 mm and a length $L$ = 150 mm was made. The compressive strength of the specimen was as follows:

$$R = P/F \tag{6}$$

where $R$ is the compressive strength of the specimen, MPa; $P$ is the damage load of the specimen, N; $F$ is the force area of the specimen, mm$^2$.

YA-300 equipment, as shown in Figure 4, was used to test the uniaxial compressive strength of the rock specimen, and the failure load measured on the rock specimen is shown in Figure 5. The uniaxial compressive strength of the prepared rock specimen was calculated to be approximately 81 MPa by substituting the peak value of the vertical coordinate of Figure 5 into Equation (6).

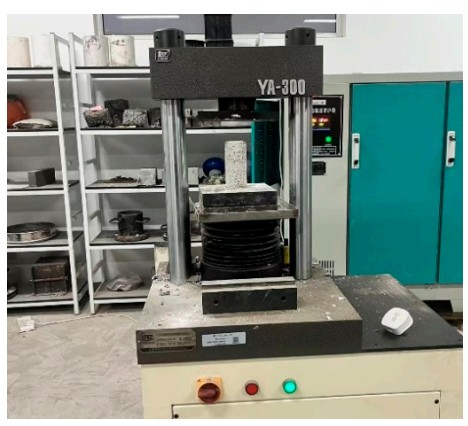
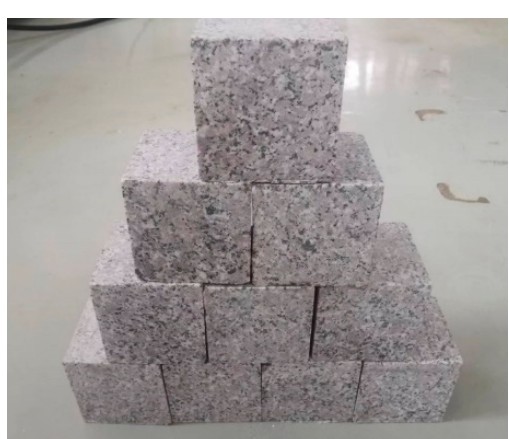

**Figure 4.** Compressive strength tester and rock specimens.

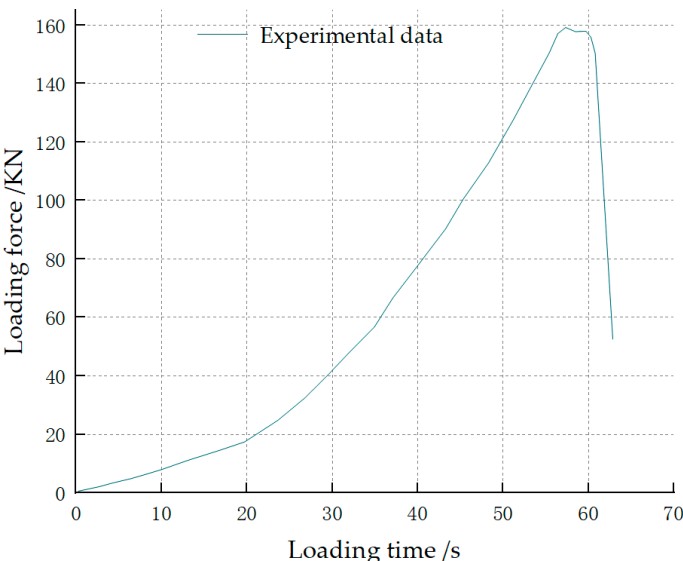

**Figure 5.** Failure load of the rock specimens.

### 3.2. Numerical Model Validation

A high-pressure twin water jet impact test was performed to confirm the accuracy of the numerical model. To ensure that the test equipment data were consistent with the numerical model, the selected nozzle diameter was 1.0 mm, the set water pressure was 50 MPa, the jet spacing was 15 mm and both water jets impacted the rock specimen vertically.

From the test pictures shown in Figure 6, the rock area between the impact points of the two water jets was damaged. The rock damage effect in this test can be compared with the numerical simulation analysis in Section 2.2 on the cause of the rock fragmentation: shear stress between the impact points of the two water jets on the rock specimen was greater than the shear strength of the rock, and the impact load between the two water jets interacted to accelerate and intensify the damage to the rock, such that cracks appeared inside the rock and converged to make the rock form a rupture crater. This research shows that the form of the damage from the high-pressure double water jets on the rock is more accurate, so the subsequent test analysis could be used to explain the damage form.

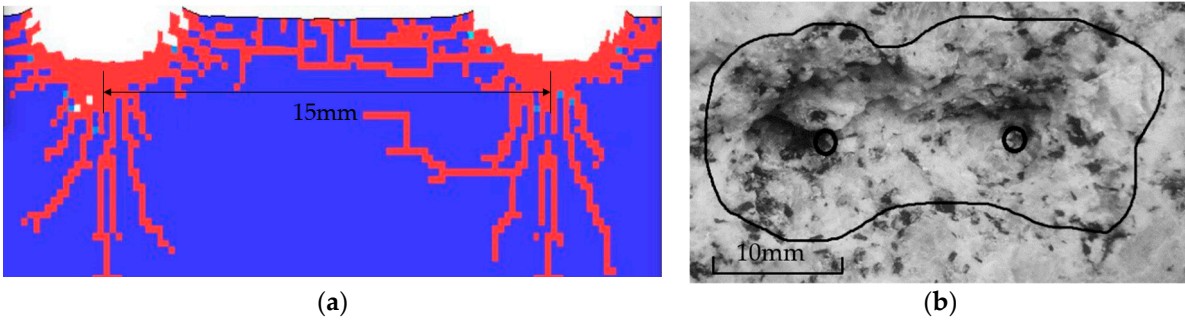

(**a**)　　　　　　　　　　　　　　　　　　　(**b**)

**Figure 6.** Result comparison verification: (**a**) numerical simulation; (**b**) experimental result.

### 3.3. Effect of Water Jet System Pressure on Rock-Breaking Effect

Three parameters were set according to the results of the numerical simulation of the high-pressure double water jet rock breaking, and the effect of a fourth parameter on the rock-breaking capacity was investigated. The rock-breaking capability of a water jet can be assessed by the volume of broken rocks. The rock-breaking volume and specific energy were used as the criteria to evaluate the rock-breaking energy of the two rock-breaking methods, with the specific rock-breaking energy of a high-pressure, dual-stream water jet and a high-pressure, single-stream water jet taken into consideration.

The method for determining the damage volume of the rock was the "equivalent water injection method" [21]. The specific procedure was to soak the rock to the water saturation state and then inject water into the broken pits with 1 mL and 5 mL medical syringes, and the volume of water injected was the damage volume of the rock. Multiple measurements were taken, and the average value was taken after eliminating the abnormal values to reduce the manual error. Since the test uses a high-pressure plunger pump as the power source, it cannot directly show the total energy consumption of a single broken rock, which needs to be calculated using Equation (7), as follows:

$$E = P \cdot Q \cdot T \tag{7}$$

where $E$ is the total energy consumption of a single broken rock, J; $P$ is the water jet pressure, MPa; $Q$ is the water jet flow rate, L/min; $T$ is the time for the water jet to impact on the rock, s. $P$, $Q$, and $T$ can be displayed using the system data.

The specific energy for rock breaking is the ratio of the total energy consumption of single rock breaking to the rock-breaking volume [22], as follows:

$$S_e = \frac{E}{V} \tag{8}$$

where $S_e$ is the specific energy for rock breaking, J/cm$^2$; $E$ is the total energy consumption of single rock breaking, J; and $V$ is the rock-breaking volume, cm$^2$.

When the water jet pressure test was conducted, the standoff distance was 3 mm, the impact time was 30 s, the impact angle was 0°, the spacing between the two water jet impact points was 1.5 cm, the water jet nozzle diameter was selected as 1 mm, and the rocks were impacted with pump pressures of 40 MPa, 42 MPa, 44 MPa, 46 MPa, 48 MPa and 50 MPa. The method was to impact the rock once with a high-pressure double water jets and twice with a high-pressure single water jet at a spacing of 1.5 cm. The two impact methods are single variable tests, and only different pressure variables are set. The two impact methods using the same pressure comprised the same group test, and five groups of tests are set up.

According to the analysis shown in Figure 7, it can be seen that the damaged areas caused by the two rock-breaking methods increased with the increase in the water jet pressure. When the pressure was 44 MPa and 42 MPa, the damage pits caused by the two methods converged, which basically accords with the results of the numerical simulation. However, with an increase in the pressure, the damaged areas caused by the double water jets is obviously larger than that by single water jet. In addition, when the pressure was greater than 44 MPa, the damage to the rock caused by the high-pressure double-stranded water jet could not be distinguished from the damage edge of the impact point.

Within the range of the test pressure, the area on the rock crushed because of the high-pressure single water jet presented an "8" or "gourd" shape, which can clearly be distinguished from the crushed edge of the single impact point. However, the fracture pit with the two water jets that impacted on the rock at the same time had a larger scope and more complete intersection. This is because when the two water jets impacted on the rock, the shear and tensile stress caused by the two water jets impacting on the rock between the impact points was greater than the stress created by the two water jets impacting on the rock, which led to the difference in the form of the damage.

Figure 8 shows the rock-breaking volumes and specific rock breaking energies of the two water jets. The trend in the volume variation of broken rock in the figure is consistent with that in Figure 7, and the specific energy of the broken rock decreased with an increase in the pressure. With an increase in the system pressure, the ratio of the rock-breaking volume of the double water jets to the single water jet increased from 1.52 to 2.11, and the ratio of the specific rock-breaking energy dropped from 0.62 to 0.51. In other words, under the same parameter settings, the rock-breaking volume and specific energy were the largest and the lowest when the system pressure was 50 MPa. In this case, the rock-breaking

volume of the high-pressure double water jets was 2.11 times that of the single water jet's two impacts, but the energy consumed was only 50% of that of the single water jet. Under similar energy consumption conditions, the rock-breaking effect of the high-pressure double water jets impinging on the rock once was better than that of the single water jet impinging on the rock twice, and the working time consumed was only half of the latter.

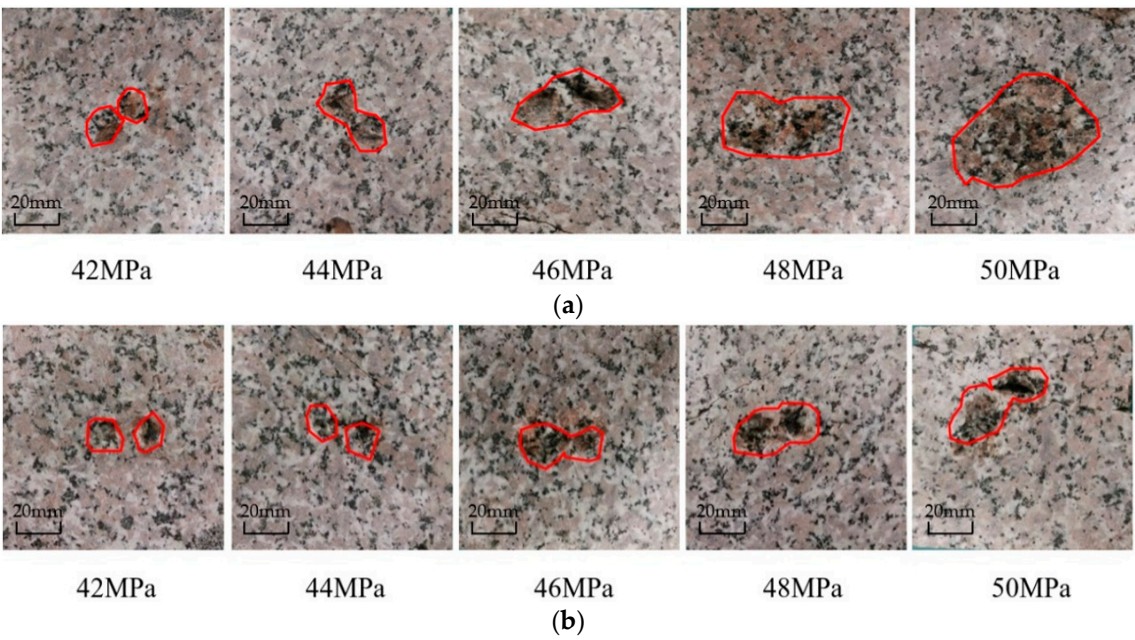

**Figure 7.** Damage forms of rocks caused by water jets with different pressures: (**a**) damage caused by double water jets impinging on the rock; (**b**) damage caused by a single water jet impinging on the rock.

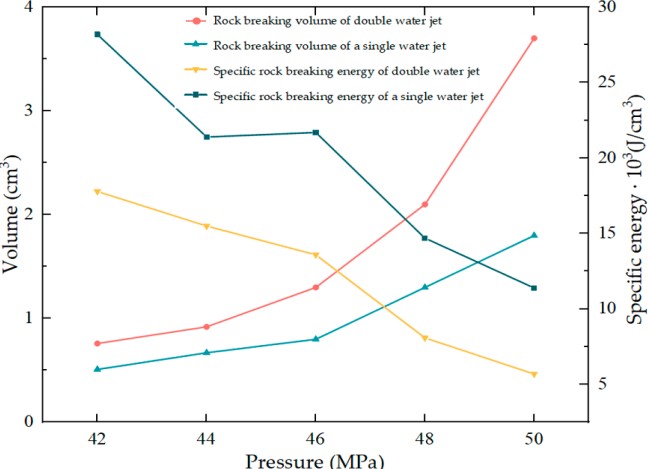

**Figure 8.** Specific rock breaking energy and volume of the water jet under different pressures.

### 3.4. Influence of Water Jet Nozzle Diameter on the Rock-Breaking Effect

In the water jet impact test with different diameters, the standoff distance used was 3 mm, the impact time was 30 s, the impact angle was 0°, the impact pressure was 50 MPa, and the distance between the two water jet impact points was 1.5 cm. The rock was impacted by nozzles with diameters of 0.8 mm, 1.0 mm, 1.2 mm, 1.4 mm and 1.6 mm. The impact mode was the same as above, and the changes in the different nozzle diameters on the broken volume of the rock specimen were calculated, and the results are shown in Figure 8.

By comparing the two sets in Figure 9a,b, it can be found that the damage effect of the high-pressure single water jet on the rock specimen was weaker than that of the high-pressure double water jet. With an increase in the diameter of the water jet, the expansion of the broken area caused by the single water jet on the rock was small, and there was an obvious "concave" shape in the middle of the damaged area of the rock. This was because the stress waves produced by two water jet impacts did not appear superimposed. The rock-breaking area of the double water jets was relatively large, and the expansion of the damaged area was relatively obvious, with a trend of initially increasing and then decreasing. During the process of increasing the water jet diameter from 0.8 mm to 1.2 mm, the rock crushing volume increased significantly. It can be intuitively found in Figure 9a that when the diameter of the water jet nozzle was 1.2 mm, the damage caused to the rock specimen was the largest. However, when the diameter of the water jet was increased from 1.2 mm to 1.4 mm, the crushing volume of the rock decreased. This is because with the increase in diameter of the water jet, the flow rate of water impacting the rock increased in unit time, and the return water generated after the water jet impacted the rock also increased with the increase in the diameter of the nozzle, thus causing interference and a weakening effect on the raw water jet. The damage to rock was reduced, so the diameter with lower energy consumption can be selected according to the actual working conditions.

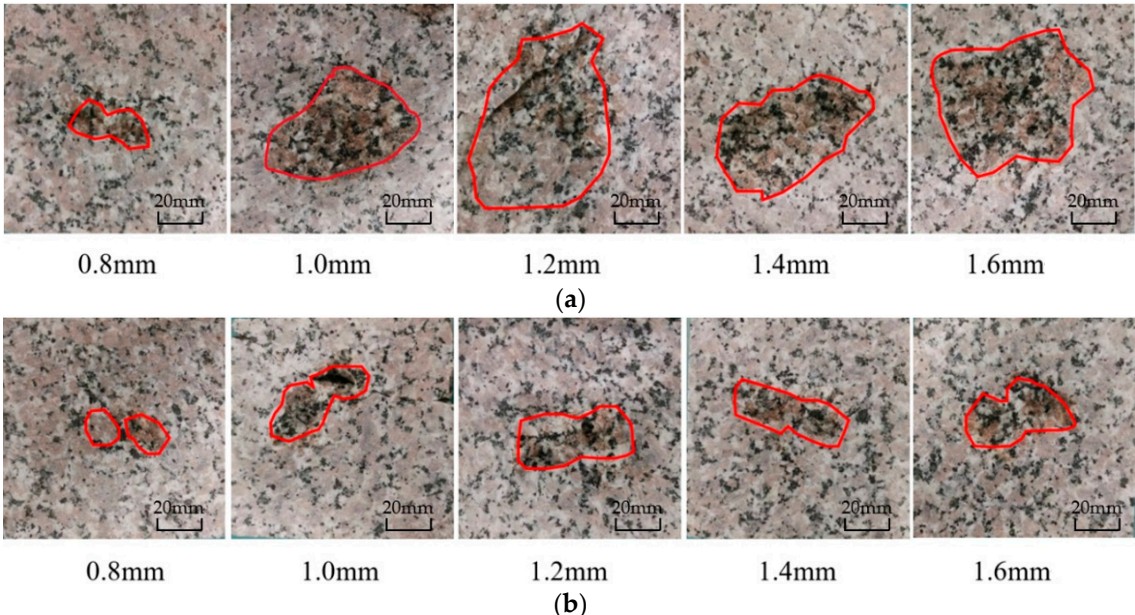

**Figure 9.** Damage patterns of rocks caused by water jets of different diameters. (**a**) Damage caused by double water jets impingement on rock. (**b**) Damage caused by single water jet impingement on rock.

Figure 10 shows the rock-breaking volumes and specific energies of rock breaking caused by two kinds of water jets. When a nozzle with the same diameter is used, the rock-breaking volume of the high-pressure double water jets is higher than that of the single water jet. When the nozzle diameters were 1.2 mm and 1.6 mm, the ratios of the rock-breaking volumes of the front and back two ways were 2.43 and 2.52, respectively. Meanwhile, it can be seen from the figure that when the nozzle with the diameter of 1.2 mm was used, the rock-breaking volumes of the two ways were 2.43 and 2.52, respectively. The rock-breaking volume of the high-pressure double water jets reached the maximum in the test value range. The curve trend in rock-crushing specific energy reached its lowest value when the nozzle diameter was 1.2 mm. In this case, the actual rock-crushing specific energy of the high-pressure double water jets was 40% of that of the single water jet, but the rock-breaking volume was 2.43 times that of the latter.

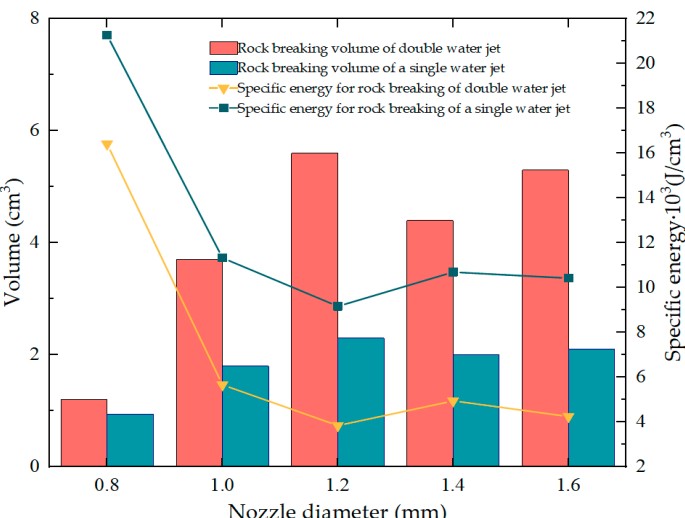

**Figure 10.** Rock-breaking volumes and specific energy consumptions of water jets with different diameters.

### 3.5. Influence of Water Jet Impact Angle on Rock-Breaking Effect

The impact angle of the water jet is also an important factor affecting its rock-breaking effect, so the included angle between the axial direction of the two water jets was defined as $\alpha$. In this section, rock-breaking tests were carried out for the impact angle of the high-pressure double-stranded water jet. The pressure of the water jet system was set as 50 MPa, the diameter of water jet nozzle was set as 1.2 mm, and the impact angles $\alpha$ were set as $0°$, $15°$, $30°$, $45°$ and $70°$. The other parameters were unchanged.

As can be seen directly from Figure 11a, when the double water jets impinged on the rock at an angle of $15°$, the damage to the rock specimen was the largest. With an increase in the impact angle, the damage area of the rock specimen changed little, but the damage depth of the rock specimen changed gradually. The damage depth of $45°$ and $70°$ was less than 2 mm. This phenomenon indicates that the high-pressure water jet only "scoured" a relatively loose layer of the rock slag on the surface of the rock specimen, but did not cause damage to the deep rock. This is because the cutting depth of the high-pressure water jet with a large impact angle was small when it impinged on rock, and the bunching property after the impact of water jet was reduced. The impact load of the water jet dissipated with the shunt generated after impingement on rock, such that the forward impact load and tangential load inside the rock could not exceed the rock failure strength. Combined with the numerical simulation results of the impact of water jet angle on rock, the water jet with an impact angle of $15°$ had the most obvious damage to rock.

The results presented in Figure 12 illustrate the rock breakage volume and specific energy for two groups of comparison tests. It is observed that when the impact angle is $15°$, the double water jets causes the most significant damage to the rock specimen, resulting in the lowest specific energy for rock crushing within this test group. Additionally, when comparing the two modes of rock breaking (double water jets and single water jet), it is found that at an impact angle of $30°$, their ratio of the rock-breaking volume reaches a minimum value of 1.54; whereas at an impact angle of $15°$, this ratio reaches its maximum value of 3.56. At an impact angle of $15°$, both the rock breakage volume and specific energy for rock breaking reach optimal values. In conclusion, when using a water jet with an impact angle of $15°$, we achieve lower specific energy for rock crushing and a larger volume of rock breakage compared to other angles tested. As the impact angle increases, however, there is a decrease in bundling properties exhibited by the water jet along with dispersed load distribution on the rock specimen leading to shallower damage depth. Therefore, it can be concluded that the impact angle significantly influences the extent of damage inflicted on rocks.

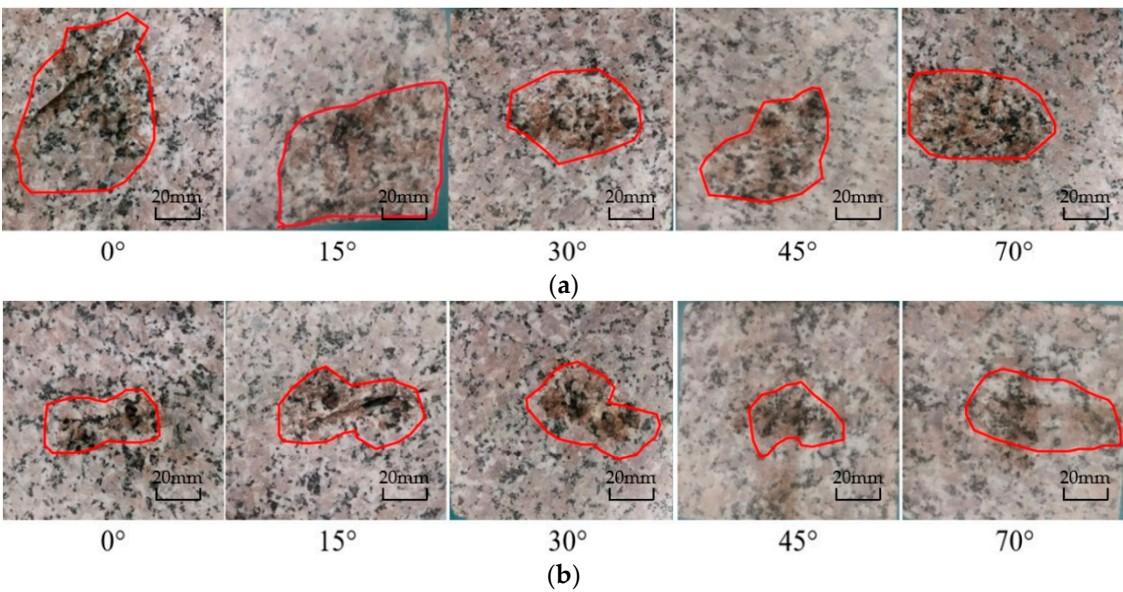

**Figure 11.** Damage morphology of rocks caused by a water jet with different angles. (**a**) Damage caused by double water jets impingement on rock. (**b**) Damage caused by single water jet impingement on rock.

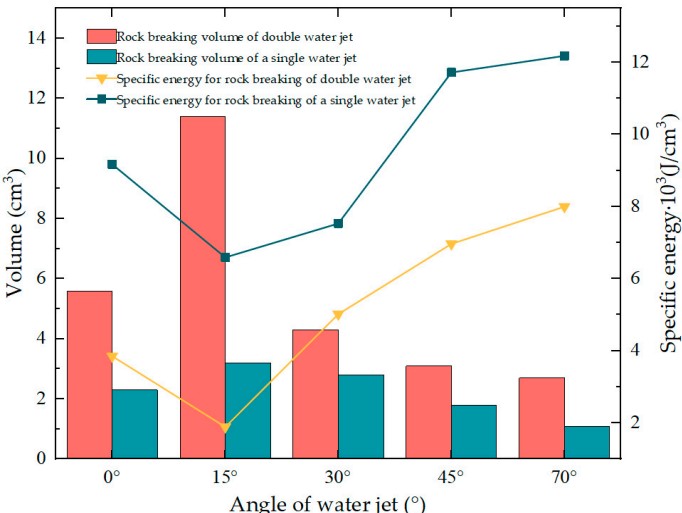

**Figure 12.** Damage morphology of rocks caused by a water jet with different angles.

### 3.6. Influence of Water Jet Impact Point Spacing on Rock-Breaking Effect

For the double-stranded water jet, the distance between impact points on the target rock is also an important factor affecting the damage volume of the rock. Therefore, in this section, only double-stranded water jets impacting on rock specimens were conducted. The system pressure was set at 50 MPa, the nozzle diameter was set at 1.2 mm and the water jet impact angle was set at 15°. Based on the minimum distance of d = 1.5 cm between two jet holes, the test impact point spacings used were 1.0 d, 1.5 d, 2.0 d, 2.5 d and 3.0 d. The results are shown in Figure 13.

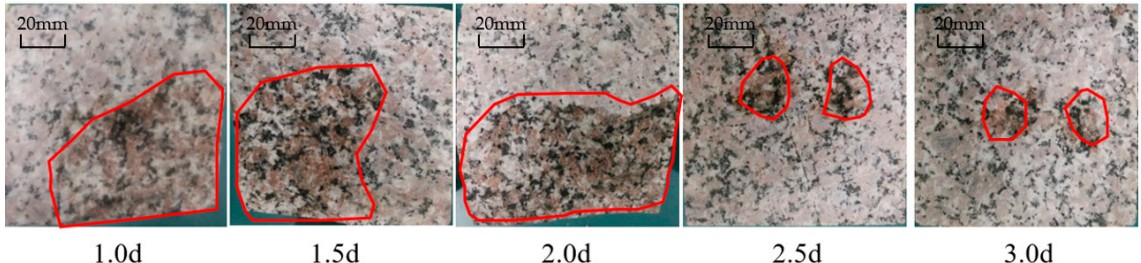

**Figure 13.** Damage form of rocks caused by water jets with different impact point spacings.

It is evident from the figure that, on the whole, the damage area initially increased and then decreased. When the distance between impact points was greater than 2.0 d, the water jet only caused two independent impact pits on the rock specimen, which is similar to the rock-breaking effect of a single water jet. This shows that the coupling and superposition effects of the impact load of the two water jets on rock and the stress waves generated in the rock greatly reduced. When the distance was greater than this, the double-stranded water jet could only cause minor internal damage to the rock area between the impact points. Because of the long distance of the stress wave transmission, it attenuated greatly, and the shear stress and tensile stress generated could not cause effective damage to the rock.

It is worth noting that penetrating fracture damage was caused to the rock at 2.0 d. The volume of broken rock at a 1.5 d spacing was already at the maximum, and as the spacing increased, the expanding cracks generated inside the rock intersected and penetrated the rock, so penetrating fracture damage occurred.

As shown in Figure 14, the rock damage volume at the 2.0 d interval was the largest, and the corresponding required energy consumption was also the lowest. The volume of rock breakage increased first and then decreased. When the impact interval was 2.0 d, the volume of the rock breakage decreased sharply because of the weakening of the stress wave transmission. In addition, high-pressure, double-stranded water jets with large spacing can also cause damage to rocks, but because of the large spacing, the crushing effect is basically consistent with the crushing volume and shape of single-stranded water jet impinged twice with the same interval.

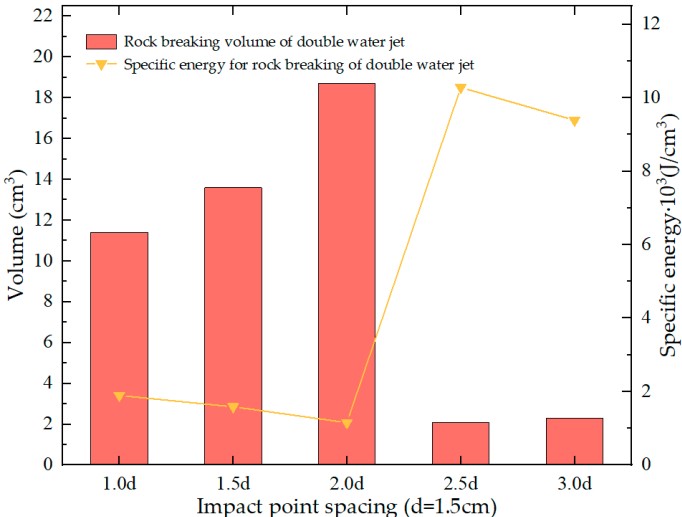

**Figure 14.** Rock-breaking volume and specific energy of water jet at different intervals.

### 3.7. Analysis of Optimal Rock Fracture Combination Parameters

In this section, the high-pressure, double-stranded water jet was mainly used to explore the influence of its various parameters on the breaking energy lithology. The parameters changed in the test were the following: water jet system pressure, nozzle diameter, impact angle and impact point spacing; the standoff distance was fixed at 3 mm.

Based on the above test results, the level values of the factors with poor effects were removed. The specific orthogonal test table is shown in Table 3. The performance evaluation index used was the rock crushing volume value of each group; the larger the value, the better the parameters of this group. Figure 15a was drawn according to the rock crushing volume value of each group of tests. Since the influence trend of each factor in rock breaking could not intuitively be obtained from the group of figures, the average value of each level in Figure 15a was classified and calculated to obtain Figure 15b and Table 4.

**Table 3.** Orthogonal test scheme of the high-pressure double water jet.

| | Orthogonal Design | | | |
|---|---|---|---|---|
| No | Pressure /MPa | Nozzle Diameter/mm | Impact Angle/° | Impact Point Spacing/d |
| 1 | 44 | 1.0 | 0 | 1.0 |
| 2 | 44 | 1.2 | 15 | 1.5 |
| 3 | 44 | 1.4 | 30 | 2.0 |
| 4 | 44 | 1.6 | 45 | 2.5 |
| 5 | 46 | 1.0 | 15 | 2.0 |
| 6 | 46 | 1.2 | 0 | 2.5 |
| 7 | 46 | 1.4 | 45 | 1.0 |
| 8 | 46 | 1.6 | 30 | 1.5 |
| 9 | 48 | 1.0 | 45 | 2.5 |
| 10 | 48 | 1.2 | 30 | 2.0 |
| 11 | 48 | 1.4 | 15 | 1.5 |
| 12 | 48 | 1.6 | 0 | 1.0 |
| 13 | 50 | 1.0 | 30 | 1.5 |
| 14 | 50 | 1.2 | 45 | 1.0 |
| 15 | 50 | 1.4 | 0 | 2.5 |
| 16 | 50 | 1.6 | 15 | 2.0 |

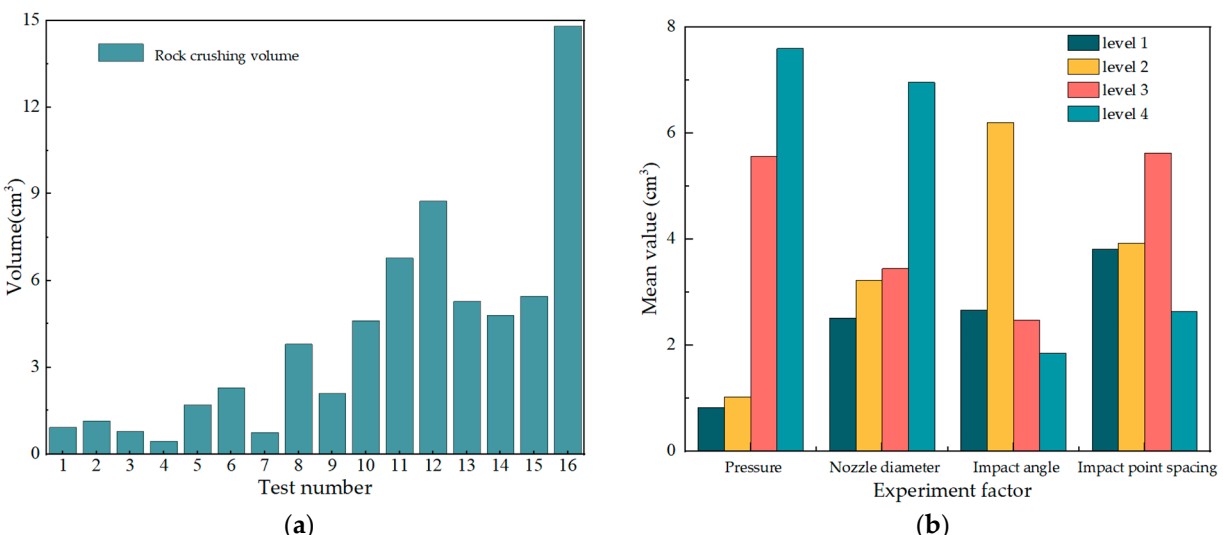

**Figure 15.** Rock-breaking volume and change trend under various factors: (**a**) rock crushing volume; (**b**) mean value.

It is evident from Figure 15b that the rock damage volume increased proportionally with an escalation in the jet pressure, while the included impact angle and distance between impact points exhibited a rising trend initially followed by a declining one during the test. In other words, there is an optimal lithologic breaking energy when the included impact angle was 15°, and the distance between impact points was 2.0 d. In the table, the range values of the jet pressure, nozzle diameter, impact angle and impact point spacing were 6.77, 3.51, 4.37 and 3.69, respectively. Therefore, the order of influence of each factor on the

high-pressure, dual-stream water jet rock breaking was A > C > D > B, namely, jet pressure, impact angle, impact point spacing and nozzle diameter. By comparing and analyzing the mean values of the rock-breaking volumes caused by each horizontal condition of the four factors, it can be concluded that the best rock-breaking combination in the test was $A_4B_4C_2D_3$, that is, the water jet system pressure was 50 MPa, the nozzle diameter was 1.6 mm, the impact angle was 15°, and the impact point spacing was 2.0 d.

**Table 4.** Range analysis of the test results.

| Factor | | Pressure | Nozzle Diameter | Impact Angle | Impact Point Spacing |
|---|---|---|---|---|---|
| Horizontal mean | x1 | 0.82 | 2.51 | 6.22 | 3.81 |
| | x2 | 1.02 | 3.22 | 2.68 | 3.92 |
| | x3 | 5.57 | 3.44 | 2.47 | 5.82 |
| | x4 | 7.59 | 6.57 | 1.84 | 2.13 |
| Range R | | 6.77 | 3.51 | 4.37 | 3.69 |
| Rank | | 1 | 4 | 2 | 3 |

## 4. Conclusions

In this research, the coupling algorithm of SPH and FEM was used to simulate the high-pressure double water jet rock-breaking process; then, the feasibility of the high-pressure double water jet rock-breaking method was verified using experiments. Furthermore, the influences of the jet pressure, nozzle diameter, jet angle and jet spacing on the rock fragmentation volume and specific rock breaking energy were analyzed based on the existing experimental apparatus. The research results are as follows.

(1) The single factor test analysis method was used to deduce the following conclusion: when the pressure of the double water jets was 50 MPa, the rock breaking volume was the largest, and the specific rock breaking energy was the smallest. Meanwhile, the rock-breaking volume of the double water jets under high pressure was 2.11 times that of single water jets under two impacts. When nozzles with the same diameter were used, the rock-breaking volume of the high-pressure double water jets was higher than that of single water jet. When the water jet impact included an angle of 15°, the specific energy for rock breaking was at its minimum and resulted in the largest volume of rock fragmentation. With an increase in the angle, the clustering property of the water jet impact decreased, and the impact load of the rock specimens was dispersed, resulting in a shallow depth of damage. Therefore, the included angle of the water jet impact has a great influence on the damage depth of rock. Under the influence of the stress wave superposition effect, the rock damage volume is the largest and the corresponding specific energy consumption is the lowest when the impact point spacing of double water jets is 2.0 d. When the impact distance is too large, the damage effect is basically consistent with the damage volume and shape of the two impacts of single water jet at the same distance.

(2) Orthogonal tests were conducted to obtain the optimal combination of factor parameters for high-pressure, double-stranded water jet rock breaking. The results show that the influence of the above factors on the lithologic breaking energy of the high-pressure, double-stranded water jet is, in the following order, jet pressure > impact angle > impact point spacing > nozzle diameter, and the optimal combination of factor parameters and performance for rock breaking is $A_4B_4C_2D_3$, that is, a water jet system pressure of 50 MPa, nozzle diameter of 1.6 mm, impact angle of 15°, and impact point spacing of 2.0 d. The above research conclusions can provide guidance and reference for improving the efficiency and economics of crushing rocks with high-pressure water jets.

There are still several deficiencies in this research. Firstly, in the numerical simulation process, the jet was assumed to be a homogeneous fluid, and the influence of initial cracks and pores on rock fractures was not considered. Second, there was no lateral cutting test on the rock. Subsequent studies will take all of the above factors into consideration to further conduct better research.

**Author Contributions:** Methodology, Y.P.; Software, S.Z.; Validation, H.Y.; Investigation, Y.P.; Data curation, K.P.; Writing—original draft, S.Z.; Writing—review and editing, F.H.; Funding acquisition, Y.P. All authors have read and agreed to the published version of the manuscript.

**Funding:** This research was supported by the Natural Science Foundation of Hebei Province Ecological Intelligent Mine Joint Fund (E2022402102). This support is greatly acknowledged and appreciated.

**Data Availability Statement:** The datasets generated and analyzed during the current study are available from the corresponding author upon reasonable request.

**Acknowledgments:** Z.Z. and L.X. are acknowledged for their valuable technical support.

**Conflicts of Interest:** The authors declare no conflict of interest.

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
