# Peer review of "Rock-Breaking Characteristics of High-Pressure, Dual-Stranded Water Jets"

_processes, doi:10.3390/pr11092562_

Round 1
Reviewer 1 Report (New Reviewer)
CONFIDENTIAL REVIEW COMMENTS
Journal: Processes (ISSN 2227-9717)
Manuscript ID: processes-2484383
Type: Article
Title: Study on rock breaking characteristics of high-pressure double-stranded water jets
I think the manuscript does include major scientific findings, and the topic is of interest. So this manuscript could be accepted for publication after the minor revision, and some suggestions to improve the manuscript and required clarifications are given below:
1. The English writing of the manuscript should be improved, and the author should pay more attention to English grammar, spelling, and sentence structure so that the goals and results of the study are clear to the readers.
2. For readers to quickly catch your contribution, it would be better to highlight major difficulties and challenges, and your original achievements to overcome them, in a clearer way in abstract and introduction.
3. The research gaps are not properly identified. It has to be incorporated by duly justifying the importance of work carried out, while revising the manuscript.
4. More details about the LS-DYNA software is necessary to explain your work.
5. It is better to include a section “Practical Importance of the Study” while revising the manuscript. In this section, authors should clearly explain how the study carried out by authors is superior to existing studies in similar lines. (OPTIONAL)
OK
Author Response
Please see the attachment.

Reviewer 2 Report (New Reviewer)
Study on rock-breaking characteristics of high-pressure double-stranded water jets
Reviewers' comments:
This paper presents a numerical study on rock-breaking characteristics of high-pressure double-stranded water jets, based on the SPH–FEM coupling algorithm. As a whole, following point needs to be addressed before it can be adaptable for publication in this journal and needs to be reviewed again.
1. Introduction part should begin with a pleasing statement devised to provoke the readers' attention and let the reader know the topic of the paper.
2. Some grammatical points or vague phrases are found that can be better exhibited. For e.g. “The above studies show that the form and mechanism……” isn’t supplied with appropriate verb, and etc.
3. Numerical simulation with relevant governing equations isn’t exposed appropriately in the manuscript and should be dataily clarified.
4. Regarding the Influence of water jet impact angle on rock-breaking effect author should give more quantitative analysis for better judgment.
5. Section 7.3 Analysis of optimal rock fracture combination parameters; should be better interpreted.
6. The conclusion seems to have prolixity parts; some quantified points should be highlighted; pointing out that the issues are drawn with the present limitations.
edition is essentially required
Author Response
Please see the attachment.

Reviewer 3 Report (New Reviewer)
1. Line spacing should be checked.
2. Abbreviations full meaning should be explained in text.
3. Lines 279-285: Authors write about ,,single water jet" and make conclusions for ,,double water jet". Authors should be more precise.
4. Impact angle of single water jet should also be investigated for means of comparision.
Minor editing should be conducted.
Round 2
Reviewer 2 Report (New Reviewer)
the quality of the paper has been improved; however it can be accepted for publication
This manuscript is a resubmission of an earlier submission. The following is a list of the peer review reports and author responses from that submission.
Round 1
Reviewer 1 Report
1.The manuscript has made some revisions, but I also noticed, some references have been deleted compared to the initial manuscript. I wonder the reason for this operation?
2.It is recommended to mark all modifications made in the manuscript and respond to revisions one by one. I noticed that some deletions or revisions were not marked.
Reviewer 2 Report
After revision, the paper can be accepted for publication
Author Response
Thank you very much for your help
Reviewer 3 Report
1. The numerical model related to geometric modeling and mechanical model is incomplete.
2. Water jets with inclined angles are difficult to achieve the desired results shown in Fig. 2.
3. ’ The impact method was to impact the rock once with a high-pressure double water jet and twice with a high-pressure single water jet at a spacing of 1.5 cm’. This conclusion lacks descriptive details to be difficult to determine its rationality.
English expression is basic clarity.
Reviewer 4 Report
The work under analysis shows how the material is removed from the point of view of the action of a double jet. Although it would have been better to combine the graphics part with the experimental tables and perhaps detailed the analysis comments between the single and the double jet, I believe that the data provided is sufficient for the audience to get a proper impression of the analysis performed.
The work under analysis shows how the material is removed from the point of view of the action of a double jet. Although it would have been better to combine the graphics part with the experimental tables and perhaps detailed the analysis comments between the single and the double jet, I believe that the data provided is sufficient for the audience to get a proper impression of the analysis performed.
Author Response
Thank you very much for your help and comments.
Reviewer 5 Report
The destruction of rocks by a jet of water has wide prospects in mining. This method of rock destruction is one of the safest. The destruction of rocks by a high-pressure water jet can be used in the open-pit mining method and in the underground mining method of mineral deposits.
The following question about this article:
It is not clear how many tests were performed for each parameter (jet pressure, impact angle, impact point spacing, nozzle diameter). If there was only 1 experiment for each parameter, then the result may be an error.
It is interesting to know what productivity of rock destruction can be achieved in an underground mine in m3/hour?
The article should be supplemented with information about direction of authors further research.
Round 2
Reviewer 1 Report
1. It was mentioned, ‘ the literature cited in the previous introduction is too old, so the introduction is modified and replaced with literature of recent years’. I think this statement is perfunctory and dishonest. If there are no incorrect citations, it is recommended to supplement the previous literature and further add some of the latest review literature in the introduction section. Meanwhile, not all modifications in the manuscript have been marked.
2.The legends in Figures 2 and 6 do not seem to match the content. Please carefully check.
3.Some units in the manuscript are not standardized. Please carefully review and revise.
4.It was mentioned, “the average stress on the contact point O between the high-pressure water jet and the rock”. What is the meaning of " point O".
5.There are obvious errors, such as, "As shown in Figure 2a, it depicts both the pure elastic stage and a plastic damage stage".
Reviewer 3 Report
1. The numerical model has not been modified or supplemented.
2. The results shown in Figure 2 are questionable.
There are a few expressions that do not conform to English expression habits.
Round 3
Reviewer 1 Report
I have made great efforts to evaluate the manuscript, but the response to the manuscript always avoids the importance.
I have made every effort and hope that other peer reviewers can better evaluate this manuscript.
Good luck!